# Escalation of Tau Accumulation after a Traumatic Brain Injury: Findings from Positron Emission Tomography

**DOI:** 10.3390/brainsci12070876

**Published:** 2022-07-01

**Authors:** Abdalla Z. Mohamed, Paul Cumming, Fatima A. Nasrallah

**Affiliations:** 1Thompson Institute, University of Sunshine Coast, Birtinya, QLD 4575, Australia; bio.abdallah2012@gmail.com; 2Queensland Brain Institute, The University of Queensland, Brisbane, QLD 4072, Australia; 3Department of Nuclear Medicine, Bern University Hospital, 3010 Bern, Switzerland; paul.k.cumming@gmail.com; 4School of Psychology and Counselling, Queensland University of Technology, Brisbane, QLD 4059, Australia

**Keywords:** traumatic brain injury, flortaucipir [^18^F]-AV1451, positron emission tomography, tau, CSF biomarkers, Alzheimer’s disease, CFS-tau, CFS-amyloid, cognitive decline

## Abstract

Traumatic brain injury (TBI) has come to be recognized as a risk factor for Alzheimer’s disease (AD), with poorly understood underlying mechanisms. We hypothesized that a history of TBI would be associated with greater tau deposition in elders with high-risk for dementia. A Groups of 20 participants with self-reported history of TBI and 100 without any such history were scanned using [^18^F]-AV1451 positron emission tomography as part of the Alzheimer’s Disease Neuroimaging Initiative (ADNI). Scans were stratified into four groups according to TBI history, and by clinical dementia rating scores into cognitively normal (CDR = 0) and those showing cognitive decline (CDR ≥ 0.5). We pursued voxel-based group comparison of [^18^F]-AV1451 uptake to identify the effect of TBI history on brain tau deposition, and for voxel-wise correlation analyses between [^18^F]-AV1451 uptake and different neuropsychological measures and cerebrospinal fluid (CSF) biomarkers. Compared to the TBI-/CDR ≥ 0.5 group, the TBI+/CDR ≥ 0.5 group showed increased tau deposition in the temporal pole, hippocampus, fusiform gyrus, and inferior and middle temporal gyri. Furthermore, the extent of tau deposition in the brain of those with TBI history positively correlated with the extent of cognitive decline, CSF-tau, and CSF-amyloid. This might suggest TBI to increase the risk for tauopathies and Alzheimer’s disease later in life.

## 1. Introduction

Traumatic brain injury (TBI) denotes brain tissue damage due to an external force [1] causing immediate tissue damage followed by secondary consequences of more complex pathologies. The secondary neuropathology of TBI includes axonal injury [2], demyelination [3,4], neuroinflammation [5,6], and aggregation of Aβ [7,8] and tau [9,10]. As such, there is considerable overlap in TBI with biomarkers of neurodegenerative disorders such as Alzheimer’s disease (AD). Previous studies have suggested an increased risk for AD in individuals with a history of TBI [3,8,9], and a four-to-five-year shift forward in the mean age of onset of clinical AD in such cases [7,11,12]. Other studies have proposed that TBI is more strongly linked to Lewy body disease or parkinsonism than with AD [13,14]. The latter findings imply a broad interaction between TBI and the subsequent vulnerability for a range of neurodegenerative diseases such as chronic traumatic encephalopathy [15,16].

Tau is a structural protein that normally binds to axonal microtubules, but can undergo abnormal phosphorylation, misfolding, and aberrant cleavage, potentially in response to a single TBI event, which may act as seeding for a “prion-like” progression of the acute tauopathy [17]. In an [^18^F]-AV1451 PET study, Robinson et al. showed elevated cerebral tau binding in Iraq and Afghanistan veterans who had suffered TBI due to blast injury; the primary foci of the tauopathy were in the cerebellum, and occipital, inferior temporal, and frontal cortical regions [18]. Another such study showed increased tau deposition in survivors of moderate–severe TBI, in whom the affected cortical regions differed between individuals, possibly due to the heterogeneity nature of the injury, with only the right occipital cortex showing a significant group-level increase in tau deposition [19]. In a previous study, we reported increased cerebral tau deposition in Vietnam War veterans with a TBI, and a more prominent increase in those TBI survivors with comorbid post-traumatic stress disorder (TBI + PTSD) [9]. In addition, the associations between tau deposition and neuropsychological measures in the TBI + PTSD group matched corresponding findings in AD, suggesting that experiencing PTSD following TBI may be a predictor for worse cognitive outcome and higher risk for developing AD later in life [9].

While epidemiological evidence indicates that TBI survivors have an increased risk of dementia in later life, neurobiological studies suggest an increased prevalence and/or severity of AD-associated pathology in younger TBI survivors with a single TBI incident [11,16,20,21,22,23], and chronic traumatic encephalopathy (CTE) in those with multiple TBI incidents including athletes [11,20,21]. The current study aimed to investigate how a history of TBI may modulate brain pathology and clinical expression in patients seemingly on the AD continuum. We hypothesised that (i) cerebral binding of the tau ligand [^18^F]-flortaucipir (TAUVID™) to positron emission tomography (PET) is higher in individuals with self-reported history of TBI many years earlier; (ii) the individual degree of tau accumulation is related to cognitive status in TBI survivors, and (iii) brain tau deposition in these individuals correlates with cerebrospinal fluid (CSF) markers of tau and amyloid.

## 2. Materials and Methods

Data used in this research were derived from the Alzheimer’s Disease Neuroimaging Initiative (ADNI) data base. ADNI was launched in 2003 as a public-private partnership, led by Principal Investigator Michael W. Weiner, MD. Its primary goal is to combine serial MRI, PET, other biomarkers, and neuropsychological assessments to characterize the progression of AD. Ethics approval of this study to use the de-identified data was obtained through the Human Research Ethics Committee at The University of Queensland, Australia (IRB number #2017000630).

### 2.1. Study Design

As of 2020, 118 participants from the ADNI database had a self-reported history of TBI, of whom only 20 had been examined with [^18^F]-AV1451 ([^18^F] flortaucipir) PET and structural T1-weighted MRI. For comparison, we randomly selected from within the database a demographically matched group of 100 participants without history of TBI who had undergone these same imaging examinations. We downloaded the imaging data from these cases, along with their scores in a battery of neuropsychological assessments (http://adni.loni.usc.edu/ (accessed on 30 July 2020)). Each participant had been examined for the APOE-ε4 genotype (ApoE-4 positive was defined as having one or two ApoE-4 alleles). Self-reported history of TBI was registered if the person reported “concussion” and/or “head injury”.

History of TBI was defined based on a retrospective record collected by the ADNI team as part of the Pre-Existing Symptoms Checklist completed at screening. As per the ADNI procedure manual, the participants were asked to provide any medical history or health issues including history of TBI. More information about the procedure to perform each of these tests is described in https://adni.loni.usc.edu/wp-content/uploads/2008/07/adni2-procedures-manual.pdf (accessed on 30 July 2020). To extract the information regarding history of TBI status, we used the documents (MEDHIST.csv and RECMHIST.csv) available upon download from the ADNI-DOD website.

The 120 participants were classified into four groups based on their self-reported history of TBI, and whether they were symptomatic (clinical dementia rating (CDR) score ≥ 0.5) or asymptomatic (CDR = 0) for cognitive decline. Thus, the groups were (1) participants with self-reported history of TBI and a CDR ≥ 0.5 (TBI+/CDR ≥ 0.5, *n* = 10) or (2) CDR = 0 (TBI+/CDR = 0, *n* = 10), and (3) participants without history of TBI and a CDR ≥ 0.5 (TBI-/CDR ≥ 0.5, *n* = 50) or (4) CDR = 0 (TBI-/CDR = 0, *n* = 50). Three participants reported having two TBI incidents, while one participant reported three TBI incidents. The time since injury was defined as the number of years between the latest reported TBI and the PET scans.

### 2.2. Cognitive Measures

The battery of cognitive and neuropsychological measures consisted of the following: Clinical Dementia Rating (CDR) [24]; Mini-Mental State Exam (MMSE) [25]; Montreal Cognitive Assessment (MOCA) [26]; Alzheimer’s Disease Assessment Scale-Cognitive (ADAS-Cog) [27]; Everyday Cognition (ECog) [28]; Geriatric Depression Scale [29]; and Functional Assessment Questionnaire (FAQ) [30]; the Clock Drawing test [31]; Clock Copy test [31]; Rey Auditory Verbal Learning test [32]; Category Fluency test [33]; Trail Making test [34]; Boston Naming test [35]; and the American National Adult Reading Test [36]. More information about the procedure to perform each of these tests is described in https://adni.loni.usc.edu/wp-content/uploads/2008/07/adni2-procedures-manual.pdf (accessed on 30 July 2020).

### 2.3. Cerebrospinal Fluid (CSF) Sample Collection and Analysis

CSF was collected through lumbar puncture using a 20- or 24-gauge spinal needle, as described in the ADNI procedures manual (http://www.adni-info.org/ (accessed on 30 July 2020)). In brief, the CSF samples were stored in polypropylene transfer tubes, frozen on dry ice within one hour of collection, shipped overnight on dry ice to the ADNI Biomarker Core laboratory, thawed and divided into aliquots (0.5 mL), labelled, and stored at −80 °C. The levels of Aβ42, total tau, and phosphorylated tau at threonine 181 (p-tau) were later measured in each of the CSF ADNI baseline aliquots using the multiplex xMAP Luminex platform (Luminex Corp, Austin, TX, USA) with Innogenetics (INNO-BIA AlzBio3; Ghent, Belgium; for research use–only reagents) immunoassay kit–based reagents.

### 2.4. MRI/PET Image Acquisition

At each imaging site, participants were scanned with the standardized ADNI MRI protocol. Quality control of the MRI data was performed at a designated MRI Center, and detailed descriptions of imaging protocols are found at http://adni.loni.usc.edu/methods/documents/mri-protocols/ (accessed on 30 July 2020). The PET scans were acquired following intravenous bolus administration of 370 MBq (10.0 mCi ± 10%) [^18^F]-AV1451, with a total of six frames (5 min/frame) acquired during the interval of 75–105 min post-injection. After decay and attenuation correction, each frame was iteratively reconstructed in 3D with a matrix = 128 × 128 × 63, FOV = 256 × 256 × 126 mm, with isotropic resolution of 2 mm.

### 2.5. Data Pre-Processing

A study-specific template was generated with Statistical Parametric Mapping (SPM12, www.fil.ion.ucl.ac.uk/spm (accessed on 30 May 2020)). For this, the individual T1-weighted MRI images were segmented into grey matter, white matter, and cerebrospinal fluid (CSF) using SPM-DARTEL [37] based on a priori anatomical templates. The segmented T1-weighted MRI images were then resampled to 1.5 mm isotropic resolution. The SPM-DARTEL pipeline was run for six iterations to produce the study specific template.

The downloaded [^18^F]-AV1451 PET images in DICOM format were pre-processed by the ADNI team as described in http://adni.loni.usc.edu/methods/pet-analysis-method/pet-analysis/ (accessed on 30 July 2020). First, the five-minute PET frames were linearly co-registered to correct for motion artefacts. These re-aligned frames were then averaged to generate the standard uptake value (SUV) maps for each participant. The SUV maps were resampled and standardised to a 160 × 160 × 96 matrix with 1.5 mm isotropic resolution. The standardised SUV maps were smoothed with a scanner-specific filter function to an isotropic resolution of 8 mm.

Next, the SUV maps were linearly co-registered to the corresponding native-space T1-weighted MRI image of each participant. To generate the referenced SUV (SUVr) maps, each individual’s SUV map was scaled to the mean activity in the cerebellar grey matter, which was designated as the reference region [38]. The cerebellum grey matter mask was defined based on the study specific template and the inverse T1-to-study-specific-template transformations, which were used to resample the standard cerebellum mask to the individuals’ native space. These grey-matter SUVr maps were normalised to the structural study-specific template using advanced normalization tools (ANTs). Here, the individual T1 images were normalized to the study-specific anatomic template, and the warp field deformation maps were used to normalize the individual PET SUVr maps to the template.

### 2.6. Regions of Interest to Estimate Braak Staging According to [^18^F]-AV1451 SUVr

Braak staging is defined by a specific pattern of tau pathology progression, which initiates in the medial temporal lobe (stage I) to eventually encompass the neocortex (stage IV) as revealed by *post-mortem* histological examinations [39,40]. In the current study, we applied methods applied previously in [9] and developed by Schwarz et al. [40] to estimate noninvasively the Braak stage using [^18^F] AV1451 SUVr measures in the entorhinal cortex, hippocampus, superior and middle temporal gyri (STG, MTG), fusiform cortex, lingual gyrus (BA17), and pericalcarine visual cortex (V1 + V2 + V3). Each of these ROIs were defined in the study specific template and then inversely registered to the individual space to calculate the SUVr in each ROI. We then calculated the mean [^18^F] AV1451 SUVr for each ROI in each hemisphere, then used the same algorithm developed by Schwarz et al. [40]. The final Braak stage was defined as the highest score (stage) between the two hemispheres.

### 2.7. Statistical Analysis

We performed a power analysis prior to the study analysis using G*Power (3.1.9.7). Based on the expected effect size of 0.75 in two tailed distribution, and sample sizes of 100 and 20 in groups 1 and 2, the power analysis revealed critical t-value of 1.403, df of 118, power of 0.95.

#### 2.7.1. Analysis of the Neuropsychological Measures

Descriptive data are presented as mean and standard deviation, unless indicated otherwise. To identify the differences in the different neuropsychological measures between the TBI+ and TBI- subgroups, we used one-way analysis of variance (ANOVA) followed by post hoc analysis using the Wilcoxon signed-rank test to compare continuous data and the Chi-Squared test to compare categorical data. Analyses were performed with R (version 3.3.1; R Foundation for Statistical Computing, Vienna, Austria). The results were corrected for multiple comparisons using the Bonferroni correction (*p* < 0.05).

#### 2.7.2. Voxel-Based Analysis of the PET Data

To examine the effect of TBI on tau accumulation, we examined the [^18^F]-AV1451 SUVr differences in the contrast between the TBI+ and TBI- groups using a voxel-based general linear model approach, followed by ANOVA with permutation tests (FSL-randomise, 1000 permutations). We then ran analysis of covariance (ANCOVA) to investigate the correlation between tau accumulation and MMSE, MOCA, and ECOG scores in participants with history of TBI using a general linear regression model through FSL-randomise. Finally, we performed an ANCOVA analysis to establish the correlation between brain-tau and CFS biomarkers in those with history of TBI. We omitted a similar correlation analysis in those without TBI history, because issue is thoroughly documented in the literature.

All statistical analyses were corrected for ApoE4 status, age, and gender, and results were corrected for multiple comparisons using family-wise error correction (*p*  <  0.05) and threshold-free cluster enhancement.

## 3. Results

### 3.1. Study Subjects, Demographics, and Neuropsychological Data

The 20 participants from the ADNI database with a reported history of TBI included 15 males and 5 females. Their median age was 76 years (range: 63–89 years) and median time since injury was 49 years (range: 7–74 years). All demographic and clinical characteristics for the individuals with a history of TBI are shown in Table 1. In addition, a total of 100 ADNI participants without any reported history of TBI were matched based on the age, gender, and education level to the TBI group.

The different groups demographics, cognitive and behavioural scores of the different control and injury groups are shown in Table 2. There were no significant differences in age or years of education between the subgroups (*p* > 0.05). The results showed increased cognitive decline both in the TBI+/CDR ≥ 0.5 and TBI-/CDR ≥ 0.5 group when compared to TBI+/CDR = 0 and TBI-/CDR = 0, respectively. In addition, compared to TBI-/CDR ≥ 0.5, the TBI+/CDR ≥ 0.5 group showed lower MMSE scores (*p* = 0.037), higher functional assessment questionnaire scores (*p* = 0.009), and longer time to complete (*p* = 0.04) and more errors of commission (*p* = 0.004) in the Trail Making Test Part B. No such significant differences were observed between the TBI+/CDR = 0 and the TBI-/CDR = 0 groups (*p* > 0.05).

### 3.2. Tau Deposition Is Increased in the TBI+ Group

In the asymptomatic groups, the TBI+ group (TBI+/CDR = 0) showed a relative increase in [^18^F]-AV1451 SUVr in the hippocampus, inferior temporal gyrus, middle temporal gyrus, superior temporal gyrus, temporal pole, transentorhinal cortex, angular gyrus, inferior parietal gyrus, superior parietal gyrus, and precunus compared to those who had no history of TBI (TBI-/CDR = 0) (*p* < 0.05, Figure 1A). Similarly, in the symptomatic groups (i.e., CDR ≥ 0.5), the TBI+/CDR ≥ 0.5 subgroup showed increased [^18^F]-AV1451 SUVr in the left hippocampus, inferior temporal gyrus, middle temporal gyrus, inferior parietal gyrus, superior occipital gyrus, middle occipital gyrus, and inferior occipital gyrus compared to the TBI-/CDR ≥ 0.5 (*p* < 0.05; Figure 1B).

### 3.3. Braak Staging of [^18^F]-AV1451 SUVr in Subjects with History of TBI

Subjects with a history of TBI exhibited patterns of tau pathology distribution in the brain that resembled *post-mortem* findings of tau pathology in AD patients ranked by Braak staging [39,40,41]. Ten of the TBI participants exhibited a uniformly low [^18^F]-AV1451 binding, with SUVr similar to that in the reference region (Braak stage 0; *n* = 10), six of the cases showed focal increases in [^18^F]-AV1451 retention in the medial temporal gyrus corresponding to Braak stages I–III, two participants were identified as Braak stage IV, and two were Braak stage V according to established criteria [40,41].

### 3.4. Correlation [^18^F]-AV1451 SUVr and Neuropsychological Scores

Figure 2 represents the correlation between [^18^F]-AV1451 SUVr and cognitive status in subjects with a reported history of TBI, including CDR = 0 and CDR ≥ 0.5. There was a positive correlation between the ECog-language sub-score and [^18^F]-AV1451 SUVr results in the superior frontal gyrus (SFG), middle frontal gyrus (MFG), inferior frontal gyrus (IFG), hippocampus, transentorhinal cortex, inferior temporal gyrus, middle temporal gyrus, superior temporal gyrus, temporal pole, lingual gyrus, fusiform gyrus, angular gyrus, insula, inferior parietal gyrus, posterior cingulate cortex (PCC), and precunus (*p* < 0.05, Figure 2A). There was a positive correlation of the ECog-memory score with [^18^F]-AV1451 SUVr in the anterior cingulate cortex (ACC), hippocampus, transentorhinal cortex, inferior temporal gyrus, middle temporal gyrus, superior temporal gyrus, temporal pole, and insula (*p* < 0.05, Figure 2B). Similar regional correlations were observed in the ECog total scores, as shown in Figure 2C (*p* < 0.05).

There was a significant negative correlation between tau deposition to [^18^F]-AV1451 SUVr and MMSE score in the TBI+ group (including CDR = 0 and CDR ≥ 0.5 subgroups) in areas such as the SFG, MFG, IFG, hippocampus, transentorhinal cortex, inferior temporal gyrus, middle temporal gyrus, superior temporal gyrus, temporal pole, lingual gyrus, fusiform gyrus, angular gyrus, insula, superior parietal gyrus, inferior parietal gyrus, PCC, ACC, and precunus (*p* < 0.05, Figure 3A). These same regions showed negative correlations between [^18^F]-AV1451 SUVr and MOCA scores (*p* < 0.05, Figure 3B).

### 3.5. Correlation between [^18^F]-AV1451 SUVr and CSF Levels of Tau and Amyloid

CSF-Aβ_42_ levels in the TBI+ group (including CDR = 0 and CDR ≥ 0.5) correlated negatively with [^18^F]-AV1451 SUVr values in the SFG, MFG, IFG, hippocampus, transentorhinal cortex, inferior temporal gyrus, middle temporal gyrus, superior temporal gyrus, temporal pole, lingual gyrus, fusiform gyrus, angular gyrus, insula, superior parietal gyrus, inferior parietal gyrus, PCC, ACC, and precunus (*p* < 0.05, Figure 4A). There were positive correlations between the CSF-pTau concentration with [^18^F]-AV1451 SUVr in SFG, MFG, IFG, hippocampus, transentorhinal cortex, inferior temporal gyrus, middle temporal gyrus, superior temporal gyrus, temporal pole, lingual gyrus, fusiform gyrus, angular gyrus, insula, superior parietal gyrus, inferior parietal gyrus, PCC, ACC, and precunus (*p* < 0.05, Figure 4B).

## 4. Discussion

This cross-sectional observational study provides for the first time compelling evidence that a history of TBI decades previously is associated with increased tau deposition to [^18^F]-AV1451 PET and associated cognitive deterioration in subjects seemingly on the AD continuum; these findings were robust after controlling for age, gender, education level, and APOE4 status. In addition, the elevated tau deposition in the participants with self-reported history of TBI correlated inversely with the current cognitive status. Of further interest is the present finding that elevated tau deposition to [^18^F]-AV1451 in TBI survivors correlated positively with higher ECog scores, and negatively with MMSE and MOCA scores, both of which indicate associated cognitive deficits. These correlations follow a similar spatial distribution to corresponding correlation findings seen in AD patients [42,43], and we likewise observed similar correlations in our previous investigation of TBI survivors with/without comorbid PTSD [9]. In the current study, we investigated the cerebral tau deposition in elderly individuals stratified as those mild cognitive deficits with cognitive complaint (CDR ≥ 0.05) and those without cognitive complaint (CDR = 0). Our findings suggest that a TBI history earlier in life may have brought a risk for increased cerebral tau deposition and progressive cognitive impairment in advanced age.

Indeed, TBI has been reported to be an important risk factor for different types of tauopathies including chronic traumatic encephalopathy [44,45,46] and AD [6,7,12,16,17,45,46]. [^18^F]-AV1451 ([^18^F] flortaucipir) binds to neurofibrillary tangles and has been validated as a biomarker in *post-mortem* brain tissue from patients with confirmed tauopathies [47,48]. The spatial extent and magnitude of [^18^F]-AV1451 binding in PET studies of AD patients correlates with their clinical status, cognitive profile, and Braak staging [40]. Present PET results showed increased tau deposition in brain regions classically manifesting AD tau-pathology, including the medial temporal gyrus and hippocampus, extending in some patients to the lateral temporal gyrus and parietal lobe, which is suggestive of Braak stage V. These results are supported by previous PET and *post-mortem* studies in TBI survivors showing increased tau deposition in a pattern overlapping with that reported in AD [5,15,49,50], whereas other studies have reported rapid onset and persistence of elevated tau binding in about one-third of TBI survivors [15,50,51]. A *post-mortem* investigation extending up to 49 years after a single TBI event showed abundant NFTs in the cingulate gyrus, superior frontal gyrus, and insular cortex, further suggesting an overlap with AD-like neuropathological features [49]. Another *post-mortem* study by Zanier and colleagues reported on tauopathy in 15 controls and 15 closed-head TBI survivors who had suffered a single moderate or severe TBI up to 18 years earlier [50]. They showed widespread hyperphosphorylated tau pathology within all cortical layers, mainly in the hippocampus and the superficial cortical layers [50]. In addition, our previous study [^18^F]-AV1451 PET study showed increased tau deposition in similar brain regions in Vietnam War veterans who had suffered moderate-to-severe TBI almost five decades previously, with more extensive tau deposition in the TBI subgroup group with comorbid PTSD [9]. Together, these various results indicate that tau pathology can occur in the aftermath of a TBI, perhaps providing a mechanistic explanation whereby a biomechanical insult might trigger self-sustained neurodegeneration.

Tau is a scaffolding protein that binds axonal microtubules together with other proteins. A TBI event can provoke tau to undergo abnormal phosphorylation and misfolding, ultimately forming NFTs [51]. This pathological cascade may be initiated by diverse factors such as axonal injury, microhaemorrhage, astrocytosis, perivascular microgliosis, and breaching of the blood–brain barrier [52]. Previous studies proposed a biophysical model to explain the elevated tau deposition following TBI, suggesting that axonal shearing may cause tau breakage due to severe mechanical strains of rapid onset, thus causing tau dissociation and aggregation [53,54]. Another study proposed that the neuropathological cascade post-TBI might follow a feed-forward mechanism initiated by acutely increased Aβ levels, causing in turn blood–brain barrier leakage and increased arterial stiffness, and thus propagating to further amyloid and tau deposition [55]. Researchers also showed increased tau aggregation in mutant tau-transgenic mice following acute and chronic TBI [56]. Another study using wild-type mice showed that severe TBI induced progressive tau pathology that spread to the hemisphere contralateral to the injury over a period of 12 months post-TBI, corresponding to half of the murine lifespan [50]. Results of the present cross-sectional observation study in long-term TBI survivors concur in suggesting that a single TBI event may have initiated an initially circumscribed tau-deposition, which in the course of time propagated to a more spatially extensive tauopathy.

In keeping with that conjecture, we observed in the present study a positive correlation between [^18^F]-AV1451 SUVr with CSF-p-tau in participants with TBI, but a negative correlation with CSF-Aβ_42_, suggesting that the pathology to be similar to Alzheimer’s pathology. Previous studies showed similar inverse relationship patterns in Alzheimer’s pathology [57,58,59], and are in line with the amyloid/tau/neurodegeneration classification scheme [19,58]. The correlations were mainly confined to the medial and inferior temporal gyrus, but extended to include the parietal gyrus in a few cases, thus matching the tauopathy pattern typical of advanced AD. These findings of increased [^18^F]-AV1451 SUVr binding are indicative cerebral tau pathology [40] also manifesting in AD patients along with increased CSF levels of tau and ptau along with reduced CSF-Aβ [58]. Furthermore, a recent study showed a positive correlation between the total cortical [^18^F]-AV1451 uptake and CSF-tau and CSF-ptau in AD cases [19].

Chief among the limitations of this study is the small number of participants with TBI history (see Table 1). Furthermore, we note that this study relies upon retrospective self-reporting of TBI, without objective clinical details; this limitation of the database necessarily imparts some uncertainty about the nature and severity of the reported injury [60]. The lack of detailed clinical information on self-reported TBI, unmedically defined severity level of the TBI, and history of medications use are also key limitations of the study, which might have contributed to the molecular profiles of these participants. The use of [^18^F]-AV1451 PET may be vulnerable to off-target binding to monoamine oxidase in astrocytes and neuromelanin in substantia nigra cells [61], such that the present findings cannot be attributed exclusively to tau-deposition. A prospective long-term study design with a more selective tau tracer (or in individuals pre-treated with diphenyl) would enable better assessment of the nature and severity of tauopathy in the aftermath of TBI. Additionally, the frequent occurrent of PTSD comorbidity triggered by the TBI event may be an additional factor in need of consideration [8,9]; we have no information about the PTSD status of individuals in the current dataset.

## 5. Conclusions

In this retrospective cross-sectional study, a self-reported history of TBI was associated with increased risk for AD development as reflected by the increased tau deposition and cognitive deficits. In addition, the study showed correlations between tau deposition, cognitive status, and CSF biomarkers. These changes suggest that TBI history may alter or fast-forward the trajectory of or vulnerability to AD or other dementias. This work will provide the evidence base for future investigations of the long-term pathology associated with TBI.

## Figures and Tables

**Figure 1 brainsci-12-00876-f001:**
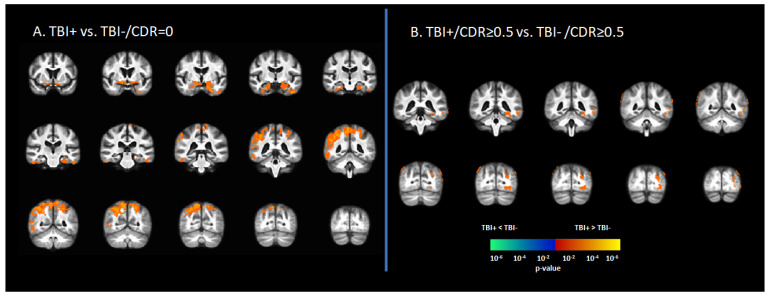
Differences in tau deposition to [^18^F]-AV1451 PET between participants with self-reported history of traumatic brain injury (TBI+) and those without TBI history (TBI-). (**A**) The statistical difference map of [^18^F]-AV1451 SUVr between the TBI+ group in contrast to the control group (TBI-/CDR = 0) group, showing that history of TBI increased the tau deposition in wide-spread brain regions. (**B**) The statistical difference map of [^18^F]-AV1451 SUVr in the symptomatic cases between the TBI+/CDR ≥ 0.5 (in contrast to TBI-/CDR ≥ 0.5) showed increased tau deposition in cortical regions overlapping with those reported for Alzheimer’s disease [40,41]. The statistical tests were corrected for nuisance covariates, including age, gender, education, and APOE-ε4 status. Red-yellow shows regions with the TBI+ group showing higher tau as compared to TBI-, while blue-green represents less tau deposition in the TBI+ as compared to TBI-.

**Figure 2 brainsci-12-00876-f002:**
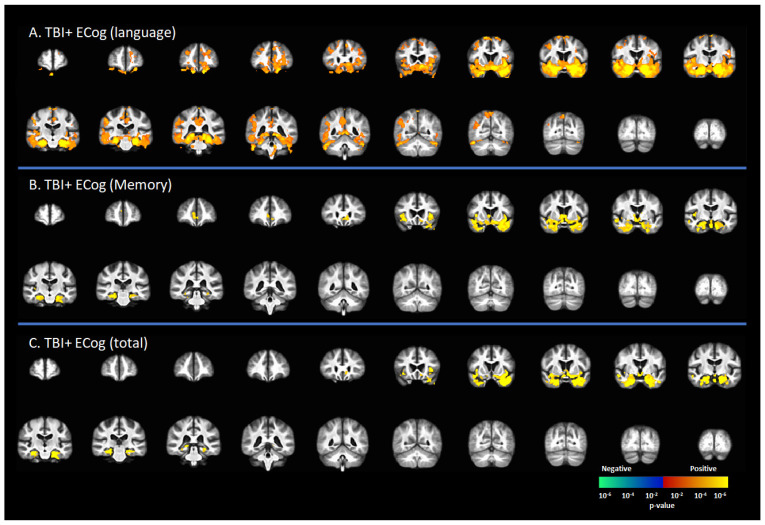
Correlation between [^18^F]-AV1451 SUVr maps and everyday cognition (ECog) scores in the population with a self-reported history of traumatic brain injury (TBI), with results of the correlations between tau deposition and (**A**) ECog-Language sub-test, (**B**) ECog-Memory sub-test, and (**C**) ECog-total score. Red-yellow represents a positive correlation between tau accumulation in people with self-reported history of TBI and ECog scores, while blue-green represents negative correlation between tau accumulation and ECog score.

**Figure 3 brainsci-12-00876-f003:**
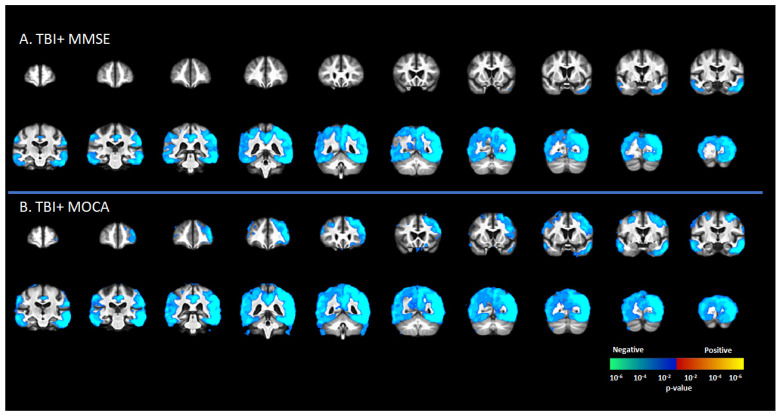
Correlation between [^18^F]-AV1451 SUVr maps and different cognitive test scores in the elderly population with a self-reported history of traumatic brain injury (TBI), with results showing negative correlations between tau deposition and (**A**) mini-mental state scale (MMSE) and (**B**) Montreal cognitive assessment (MOCA). Red-yellow represents a positive correlation between tau accumulation in people with self-reported history of TBI and cognitive scores, while blue-green represents negative correlation between tau accumulation and cognitive scores.

**Figure 4 brainsci-12-00876-f004:**
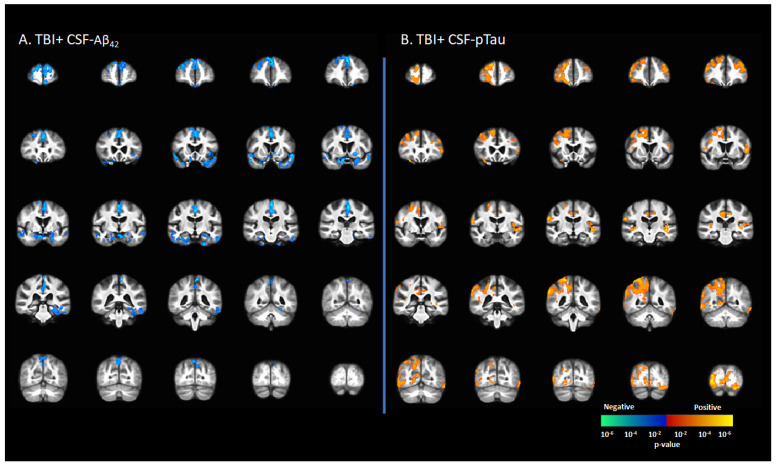
Correlation between [^18^F]-AV1451 SUVr maps and cerebrospinal fluid (CSF) biomarkers in the population with a self-reported history of traumatic brain injury (TBI), with results showing (**A**) negative correlations between CSF-Aβ_42_ and [^18^F]-AV1451 SUVr in brain; (**B**) positive correlation between CSF-pTau and [^18^F]-AV1451 SUVr in the brain. Red-yellow represents a positive correlation between tau accumulation in TBI+ cases and CSF biomarkers, while blue-green represents negative correlation between tau accumulation and CSF biomarkers.

**Table 1 brainsci-12-00876-t001:** Clinical characteristics of participants with self-reported history of TBI.

	Sex	APOE	Age	TSI	Number of TBI Incidents	Source	LOC	Braak Stage	Aβ+	CDR	MOCA	ADAS	ADAS-Cog	MMSE
P1	M	4/4	77	69	1			1	1	0	NA	5	7	29
P2	M	3/3	79	28	1	Fall	LOC	1	0	0	25	9	13	27
P3	F	2/3	74	56	1		LOC	3	1	0	26	8	11	29
P4	M	2/3	68	53	1			4	0	0	26	15	20	29
P5	M	3/3	72	9	1		LOC	0	0	0	27	8	10	29
P6	F	3/4	63	50	1			1	1	0	28	6	9	29
P7	F	3/3	83	48	1			0	0	0	27	8	13	28
P8	F	3/3	71	9	1			0	0	0	30	9	11	30
P9	M	3/4	67	54	1	Fall	LOC	0	1	0	26	6	11	29
P10	F	3/3	83	52	1		LOC	0	1	0	26	10	15	27
P11	M	3/3	85	73	1		LOC	0	0	0.5	25	12	19	29
P12	M	3/3	75	30	1	Accident		5	1	1	14	18	27	24
P13	M	4/4	81	7	2	MVA		2	1	0.5	17	15	25	24
P14	M	3/3	80	15	1	Fall	LOC	0	1	0.5	27	5	8	27
P15	M	4/4	66	52	2		LOC	5	1	0.5	21	13	23	26
P16	M	3/4	74	36	1	Fall		0	0	0.5	22	11	14	29
P17	M	2/3	68	52	3	Football	LOC	4	0	1	25	16	24	23
P18	M	3/4	83	74	1			2	1	0.5	18	19	28	22
P19	M	3/3	89	24	2			0	1	0.5	24	11	16	30
P20	M	3/4	82	8	1			0	1	1	21	12	16	30

ADAS; Alzheimer’s Disease Assessment Score; APOE, apolipoprotein E genotype; Aβ+, amyloid positive scans; LOC, loss of consciousness; CDR, Clinical Dementia Rating; MMSE, Mini-Mental State Examination; MOCA, Montreal Cognitive Assessment; NA, not available; TSI, Time Since Injury (years); TBI, traumatic brain injury.

**Table 2 brainsci-12-00876-t002:** Groups demographics and cognitive performance.

	TBI-/CDR ≥ 0.5	TBI-/CDR = 0	TBI+/CDR ≥ 0.5	TBI+/CDR = 0	TBI-/CDR ≥ 0.5 vs. TBI-/CDR = 0	TBI-/CDR ≥ 0.5 vs. TBI+/CDR ≥ 0.5	TBI-/CDR = 0 vs. TBI+/CDR = 0	TBI+/CDR ≥ 0.5 vs. TBI+/CDR = 0
**number of participants**	*N* = 50	*N* = 50	*N* = 10	*N* = 10				
**Gender:**					1	0.01	1	0.065
** Female**	24 (48.0%)	25 (50.0%)	0 (0.00%)	5 (50.0%)				
** Male**	26 (52.0%)	25 (50.0%)	10 (100%)	5 (50.0%)				
**APOE4 Status:**					1	1	1	1
** Negative**	30 (60.0%)	32 (64.0%)	5 (50.0%)	7 (70.0%)				
** Positive**	20 (40.0%)	18 (36.0%)	5 (50.0%)	3 (30.0%)				
**Age**	75.1 (7.82)	76.8 (6.30)	76.6 (7.63)	75.4 (7.31)	0.61	0.93	0.94	0.982
**TBI Type:**					.	.	.	0.65
** Concussion**			7 (70.0%)	5 (50.0%)				
** Head Injury**			3 (30.0%)	5 (50.0%)				
**TBI Source:**					.	.	.	1
** Accident**			1 (20.0%)	0 (0.00%)				
** Fall**			2 (40.0%)	2 (100%)				
** Football**			1 (20.0%)	0 (0.00%)				
** MVA**			1 (20.0%)	0 (0.00%)				
**ADAS Total**	10.2 (3.96)	8.74 (3.68)	13.5 (4.01)	8.07 (2.01)	0.217	0.05 *	0.95	0.007 **
**ADAS-Cog**	16.0 (6.48)	12.7 (5.89)	20.1 (6.34)	11.9 (3.32)	0.037	0.2	0.98	0.01 **
**MOCA**	23.9 (3.49)	25.8 (2.92)	21.5 (4.20)	26.7 (1.58)	0.022	0.15	0.87	0.004 **
**MMSE**	28.2 (2.06)	28.9 (1.47)	26.4 (3.03)	28.6 (0.97)	0.182	0.03 *	0.96	0.047 *
**GD Total**	1.84 (2.16)	0.84 (0.96)	2.50 (2.12)	0.78 (0.83)	0.017	0.66	1	0.12
**CDR Memory**	0.53 (0.12)	0.00 (0.00)	0.65 (0.58)	0.00 (0.00)	0	0.25	0.84	<0.001 ***
**CDR GLOBAL**	0.50 (0.00)	0.00 (0.00)	0.50 (0.33)	0.00 (0.00)	0	1	1	<0.001 ***
**Every Day Cognitive Assessment**								
** Memory**	2.35 (0.72)	1.62 (0.50)	2.20 (0.72)	1.71 (0.57)	<0.001	0.89	0.98	0.33
** Language**	2.02 (0.70)	1.45 (0.41)	1.91 (0.71)	1.32 (0.31)	<0.001	0.95	0.92	0.12
** Visual spatial**	1.49 (0.59)	1.11 (0.20)	1.50 (0.56)	1.06 (0.12)	<0.001	1	0.99	0.13
** Planning**	1.57 (0.63)	1.12 (0.26)	1.54 (0.57)	1.11 (0.23)	<0.001	0.99	1	0.21
** Organization**	1.75 (0.79)	1.15 (0.24)	1.58 (0.50)	1.24 (0.52)	<0.001	0.84	0.97	0.56
** Divided Attention**	2.19 (0.86)	1.45 (0.56)	1.82 (0.75)	1.25 (0.25)	<0.001	0.44	0.87	0.29
** Total**	1.90 (0.58)	1.32 (0.26)	1.76 (0.49)	1.28 (0.28)	<0.001	0.81	0.99	0.09
**FAQ Total**	2.52 (3.80)	0.14 (0.50)	5.90 (5.99)	0.25 (0.46)	0.001	0.009 **	1	0.001 ***
**Clock drawing tests**	4.52 (1.01)	4.78 (0.51)	4.30 (1.06)	4.60 (0.70)	0.394	0.86	0.92	0.84
**Clock copy test**	4.86 (0.50)	4.82 (0.48)	4.70 (0.67)	4.50 (1.58)	0.992	0.90	0.50	0.91
**Logic memory—Story**	11.3 (4.74)	15.2 (3.81)	10.1 (4.04)	15.3 (2.54)	<0.001	0.83	1	0.03
**Logic memory- Delayed Recall**	9.40 (4.88)	14.0 (3.37)	6.80 (3.94)	15.1 (3.51)	<0.001	0.27	0.88	<0.001 ***
**Category Fluency Test**	18.4 (5.19)	21.7 (4.56)	15.6 (5.08)	20.2 (4.37)	0.005	0.36	0.80	0.16
**Trail Making Test**								
** Part A—Time to Complete (sec)**	39.1 (25.3)	31.0 (8.16)	40.1 (12.1)	31.1 (9.48)	0.113	1	1	0.69
** Part A—Errors of Commission**	0.12 (0.39)	0.12 (0.39)	0.11 (0.33)	0.00 (0.00)	1	1	0.77	0.91
** Part A—Errors of Omission**	0.26 (1.84)	0.00 (0.00)	0.00 (0.00)	0.00 (0.00)	0.707	0.93	1	1
** Part B—Time to Complete (sec)**	103 (66.7)	68.6 (27.8)	155 (90.0)	74.4 (27.4)	0.011	0.04 *	0.99	0.008 **
** Part B—Errors of Commission**	0.69 (1.06)	0.37 (0.64)	1.89 (1.76)	0.20 (0.42)	0.32	0.004 **	0.96	0.001 **
** Part B—Errors of Omission**	0.35 (1.55)	0.02 (0.14)	1.40 (2.32)	0.00 (0.00)	0.536	0.06	1	0.05 *
**Trail Making Test**								
** Forgetting**	3.47 (5.75)	3.88 (2.85)	5.33 (2.50)	4.29 (3.45)	0.975	0.76	1	0.97
** Immediate**	36.6 (11.0)	44.6 (10.5)	28.2 (6.59)	48.6 (7.52)	0.005	0.26	0.79	0.004 ***
** Learning**	5.18 (3.41)	5.81 (2.26)	2.83 (1.94)	6.29 (1.80)	0.737	0.22	0.97	0.11
** Percent Forgetting**	41.6 (80.8)	36.5 (33.4)	79.0 (23.6)	36.6 (31.1)	0.978	0.45	1	0.54
** Recognition Score**	11.2 (2.61)	12.6 (3.12)	9.40 (4.35)	13.7 (0.87)	0.072	0.29	0.77	0.01 **

TBI+, participants with self-reported traumatic brain injury; TBI−,participants without self-reported traumatic brain injury; CDR, Clinical Dementia Rating; TBI+/CDR = 0, TBI + participants with CDR  =  0; TBI−/CDR = 0, TBI− participants with CDR  =  0; TBI+/CDR ≥ 0.5,TBI + participants with CDR  ≥  0.5; TBI−/CDR ≥ 0.5, TBI− participants with CDR  ≥  0.5; ADAS, Alzheimer’s Disease Assessment Scale; Ecog, Every Day Cognitive; FAQ, Functional Assessment Questionnaire; GD Total, Geriatric Depression Scale; MMSE, Mini-Mental State Exam; MOCA, Montreal Cognitive Assessment. *, *p* ≤ 0.05; **, *p* ≤ 0.01; ***, *p* ≤ 0.001.

## Data Availability

The original data used in this project and that support the findings of this study are openly available through (https://ida.loni.usc.edu (accessed on 30 July 2020), upon approval from the ADNI project administration. The different software packages used in the preparation of this manuscript are publicly available. In addition, the codes used for the analysis of the data are available upon request from the corresponding author.

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
