# Peer review of "Escalation of Tau Accumulation after a Traumatic Brain Injury: Findings from Positron Emission Tomography"

_brainsci, 2022, doi:10.3390/brainsci12070876_

Round 1

Reviewer 1 Report

The current manuscript attempts to utilize positron emission tomography with association of clinical dementia rating scores (CDR) of cognitively normal and cognitive decline in those with and without the Hx of TBI. 

1) The authors have left formatting instructions in several places. This must be removed (intro/patents). 

2) The authors need to provide more information into the self-reporting of TBI. 
2a) Did the authors utilize the OSU-TBI ID? 
Corrigan, J.D.; Bogner, J. Initial Reliability and Validity of the Ohio State University TBI Identification Method. J. Head Trauma Rehabil. 200722, 318–329.
2b) When utilizing a self-report, the definition of TBI is critical. What definition of TBI/severity did the authors use? ex. ACRM
American Congress of Rehabilitation Medicine. Brain Injury Interdisciplinary Special Interest Group, Mild Traumatic Brain Injury Task Force Definition of mild traumatic brain injury. J. Head Trauma Rehabil. 19938, 86–87.

3) Table 1 must be revised. The authors should remove or reclassify "type". The difference between "concussion" and "head injury" is unclear. Recent literature has begun to consider the effects of even subconcussive exposures and their impact on TBI-like sequela. 

3a) The authors need to provide and statistical examine the differences of TBI severity between TBI+/CDR groups. The severity of injury can have long lasting molecular effects. 

3b) The authors need to report severity of most recent and most severe lifetime TBI for participants. 

4) The authors need to provide a power analysis demonstrating they meet the minimum requirement for statistical insight. 

5) The authors need to address the limitations of their investigation, particularly regarding the self-report of TBI, and severity of TBI/lasting effects on molecular profiles. 

Author Response

Dear Journal Editor and Reviewers,

We thank the reviewers for their valuable suggestions. We have addressed the comments related to our manuscript entitled “Escalation of tau accumulation after a traumatic brain injury: findings from positron emission tomography” and highlighted the manuscript's corresponding modifications.

Review 1

Comments and Suggestions for Authors

The current manuscript attempts to utilize positron emission tomography with association of clinical dementia rating scores (CDR) of cognitively normal and cognitive decline in those with and without the Hx of TBI. 

1) The authors have left formatting instructions in several places. This must be removed (intro/patents). 

We have checked the standardized formatting throughout the revised manuscript and removed any instructions in the clean version of the manuscript.

2) The authors need to provide more information into the self-reporting of TBI. 
2a) Did the authors utilize the OSU-TBI ID? Corrigan, J.D.; Bogner, J. Initial Reliability and Validity of the Ohio State University TBI Identification Method. J. Head Trauma Rehabil. 2007, 22, 318–329.

2b) When utilizing a self-report, the definition of TBI is critical. What definition of TBI/severity did the authors use? ex. ACRM American Congress of Rehabilitation Medicine. Brain Injury Interdisciplinary Special Interest Group, Mild Traumatic Brain Injury Task Force Definition of mild traumatic brain injury. J. Head Trauma Rehabil. 1993, 8, 86–87.

We appreciate the importance of this point that the reviewer has raised, however, we are bound  by the information provided by the consortium that generated this  large and well-established database. Therefore, we have added the below to the limitations section:

Furthermore, we note that this study relies upon retrospective self-reporting of TBI, without objective clinical details; this limitation of the database necessarily imparts some uncertainty  about the nature and severity of the reported injury [1]. The lack of detailed clinical information on self-reported TBI, unmedically-defined severity level of the TBI, and  history of medications use are also key limitations of the study, which might have contributed to the molecular profiles of these participants.”

2a) To extract the TBI status, we used documents (MEDHIST.csv and RECMHIST.csv ) that is available upon download from the ADNI-DOD website.

2b) We have used a retrospective record collected by the ADNI team as part of their Pre-Existing Symptoms Checklist completed at screening. As per the ADNI procedure manual, the participants were asked to provide any medical history or health issues, including history of TBI. More information about the procedure to perform each of these tests is described in https://adni.loni.usc.edu/wp-content/uploads/2008/07/adni2-procedures-manual.pdf.

We added the following to the manuscript, in the methods section,

History of TBI was defined based on a retrospective record collected by the ADNI team as part of the Pre-Existing Symptoms Checklist completed at screening. As per the ADNI procedure manual, the participants were asked to provide any medical history or health issues, including history of TBI. More information about the procedure to perform each of these tests is described in https://adni.loni.usc.edu/wp-content/uploads/2008/07/adni2-procedures-manual.pdf. To extract the information regarding history of TBI status, we used the documents (MEDHIST.csv and RECMHIST.csv ) available upon download from the ADNI-DOD website.”

3) Table 1 must be revised. The authors should remove or reclassify "type". The difference between "concussion" and "head injury" is unclear. Recent literature has begun to consider the effects of even subconcussive exposures and their impact on TBI-like sequela. 
We have followed your advice and removed the column mentioned.

3a) The authors need to provide and statistical examine the differences of TBI severity between TBI+/CDR groups. The severity of injury can have long lasting molecular effects. 

This issue is addressed in Table 2, which shows no significant differences between the TBI subgroups (p=0.65).

3b) The authors need to report severity of most recent and most severe lifetime TBI for participants. 
We used the severity of the most recent reported incident as our indicator of the type of TBI injury. We also removed the extraneous column from Table 1, as advised by the reviewer.

4) The authors need to provide a power analysis demonstrating they meet the minimum requirement for statistical insight. 

We performed a power analysis for sample size calculation prior to conducting the study analyses. This is now stated in the revised section 2.7, Statistical Analysis: “We performed a power analysis prior to the study analysis using G*Power (3.1.9.7). Based on the expected effect size of 0.75 in two tailed distribution, and sample sizes of 100 and 20 in groups 1 and 2, the power analysis revealed critical t -value of 1.403, and df of 118, for a power of 0.95.

5) The authors need to address the limitations of their investigation, particularly regarding the self-report of TBI, and severity of TBI/lasting effects on molecular profiles. 

We added the following to the last paragraph of our discussion “Furthermore, the lack of detailed clinical information on self-reported TBI, unmedically-defined severity level of the TBI, or their history of medications use are also key limitations of the study, which might have contributed to the molecular profiles of these participants.”

References:

  1. Corrigan, J.D.; Bogner, J. Initial Reliability and Validity of the Ohio State University TBI Identification Method. J. Head Trauma Rehabil. 2007, 22, 318–329, doi:10.1097/01.HTR.0000300227.67748.77.

Reviewer 2 Report

1. It is advised to thoroughly revise the manuscript some structures are erroneously described. E.g. first paragraph.

2. The authors should avoid grouped references.

E.g. [9–15] (line 58); [6,11–13,19,20] (line 78).

3. The ‘‘results’’ are appearing before ‘‘methods’’.

Simera I, Moher D, Hoey J, Schulz KF, Altman DG. The EQUATOR Network and reporting guidelines: Helping to achieve high standards in reporting health research studies. Maturitas. 2009 May 20;63(1):4-6. DOI: 10.1016/j.maturitas.2009.03.011. Epub 2009 Apr 15. PMID: 19372017.

4. The IRB number should be specific to the study being reported. The present manuscript IRB number ‘2017000630’. The manuscript published in 2019 ‘2017000630’. Mohamed AZ, Cumming P, Götz J, Nasrallah F; Department of Defense Alzheimer’s Disease Neuroimaging Initiative. Tauopathy in veterans with long-term posttraumatic stress disorder and traumatic brain injury. Eur J Nucl Med Mol Imaging. 2019 May;46(5):1139-1151. DOI: 10.1007/s00259-018-4241-7. Epub 2019 Jan 7. PMID: 30617964; PMCID: PMC6451714.

5. The cognitive assessment was interesting and extensive. Why do the authors believe that significant results were only obtained with CDR? Moreover, Why did the authors divide into CDR≥0.5 and CDR=0? The reviewer would also like to request who performed the scales and if copyright permission was obtained before their use. It is advised to upload as unpublished material the copyrights.

6. A small description of each test performed should be done. It is advised to provide this data as supplementary material.

7. Statistical analysis.

a. Data distribution should be described.

b. How do the authors calculate the power of the study?

c. What was the statistical model used as well as the variables were chosen?

8. It is advised to include ‘‘*’’ when significant results are obtained in the table. The reviewer was not able to find in the manuscript the degrees of freedom as well as the confidence intervals.

9. A table with only the results of logistic regression should be provided.

10. What were the exclusion criteria? Were the individuals in the use of any medication?

Author Response

Dear Journal Editor and Reviewers,

We thank the reviewers for their valuable suggestions. We have addressed the comments related to our manuscript entitled “Escalation of tau accumulation after a traumatic brain injury: findings from positron emission tomography” and highlighted the manuscript's corresponding modifications.

Review 2:

Comments and Suggestions for Authors

  1. It is advised to thoroughly revise the manuscript some structures are erroneously described. E.g. first paragraph.

We regret these errors, which are now corrected.

  1. The authors should avoid grouped references.

E.g. [9–15] (line 58); [6,11–13,19,20] (line 78).

In revising the manuscript, we endeavored to correct the grouped references. However, sometimes the references are indeed concurring on the relevant point, thus calling for grouped citations.

  1. The ‘‘results’’ are appearing before ‘‘methods’’.

Simera I, Moher D, Hoey J, Schulz KF, Altman DG. The EQUATOR Network and reporting guidelines: Helping to achieve high standards in reporting health research studies. Maturitas. 2009 May 20;63(1):4-6. DOI: 10.1016/j.maturitas.2009.03.011. Epub 2009 Apr 15. PMID: 19372017.

We corrected the erroneous order of presentation to follow the journal format.

  1. The IRB number should be specific to the study being reported. The present manuscript IRB number ‘2017000630’. The manuscript published in 2019 ‘2017000630’. Mohamed AZ, Cumming P, Götz J, Nasrallah F; Department of Defense Alzheimer’s Disease Neuroimaging Initiative. Tauopathy in veterans with long-term posttraumatic stress disorder and traumatic brain injury. Eur J Nucl Med Mol Imaging. 2019 May;46(5):1139-1151. DOI: 10.1007/s00259-018-4241-7. Epub 2019 Jan 7. PMID: 30617964; PMCID: PMC6451714.

Indeed, the IRB should be specific to the particular study. We have outlined several studies in our IRB application to investigate the data from both ADNI and ADNI-DOD databases. In addition, in our IRB, we have outlined the current analysis as one of our intended studies that we will perform. Therefore, we believe it is entirely valid to use this IRB number for the current study.

  1. The cognitive assessment was interesting and extensive. Why do the authors believe that significant results were only obtained with CDR? Moreover, Why did the authors divide into CDR≥0.5 and CDR=0? The reviewer would also like to request who performed the scales and if copyright permission was obtained before their use. It is advised to upload as unpublished material the copyrights.

We used CDR as our criteria to differentiate and classify participants according to presence or absence of cognitive and memory problems. We used these criteria as they provide a clearer cut-off threshold as compared to other cognitive scores. In addition, we used these same criteria in our previous publication on a related topic [1]. The CDR score of 0 means no cognitive or memory complaints, while CDR ≥ 0.05 means that the person has begun to experience cognitive and memory decline to an extent often associated with an onset of dementia. By this means, we stratified the participants into a control group without any memory decline (CDR = 0) and a group with memory decline (CDR ≥ 0.05).

All data collection was performed by the ADNI team, and we are using the data in a de-identified format, as is standard for the ADNI consortium. In availing ourselves of the database, we had no contact with any of the participants in the study. All cognitive and memory scales are components of the broader ADNI project, and are widely used by researchers and clinicians as instruments for the diagnosis of dementia and other cognitive disturbances.

  1. A small description of each test performed should be done. It is advised to provide this data as supplementary material.

Asa noted above, we availed ourselves of the ADNI database, which has so far yielded several thousand peer-reviewed publications. However, we have now added the corresponding references for each test. In addition, we provide a link to the ADNI procedure manual explaining how each test was performed.

Now section 2.2. Cognitive Measures reads as follows:

“The battery of cognitive and neuropsychological measures consisted of the following: Clinical Dementia Rating (CDR) [2]; Mini-Mental State Exam (MMSE) [3]; Montreal Cognitive Assessment (MOCA) [4]; Alzheimer’s Disease Assessment Scale-Cognitive (ADAS-Cog) [5]; Everyday Cognition (ECog) [6]; Geriatric Depression Scale [7]; and Functional Assessment Questionnaire (FAQ) [8]; the Clock Drawing test [9]; Clock Copy test [9]; Rey Auditory Verbal Learning test [10]; Category Fluency test [11]; Trail Making test [12]; Boston Naming test [13]; and the American National Adult Reading Test [14]. More information about the procedure to perform each of these tests is described in https://adni.loni.usc.edu/wp-content/uploads/2008/07/adni2-procedures-manual.pdf.”

  1. Statistical analysis.
  2. Data distribution should be described.

The group numbers of participants are specified in the revised section 2.1.

“The 120 participants were classified into four groups based on their self-reported history of TBI, and whether they were symptomatic (clinical dementia rating (CDR) score 0.5) or asymptomatic (CDR=0) for cognitive decline. Thus, the groups were (1) participants with self-reported history of TBI and a CDR 0.5 (TBI+/CDR0.5, n = 10) or (2) CDR = 0 (TBI+/CDR=0, n = 10), and (3) participants without history of TBI and a CDR 0.5 (TBI-/CDR0.5, n = 50) or (4) CDR = 0 (TBI-/CDR=0, n = 50).”

  1. How do the authors calculate the power of the study?

As noted above, and as now specified in the revised manuscript, we have specified our use of the G*Power program.

  1. What was the statistical model used as well as the variables were chosen?

We have changed the statistical analysis section to read as follows:

“To examine the effect of TBI on tau accumulation, we examined the [18F]-AV1451 SUVr differences in the contrast between the TBI+ and TBI- groups using a voxel-based general linear model approach, followed by ANOVA with permutation tests (FSL-randomise, 1000 permutations). We then ran analysis of covariance (ANCOVA) to investigate the correlation between tau accumulation and MMSE, MOCA, and ECOG scores in participants with history of TBI using a general linear regression model through FSL-randomise. Finally, we performed an ANCOVA analysis to establish the correlation between brain-tau and CFS biomarkers in those with history of TBI. We omitted a similar correlation analysis in those without TBI history, because issue is thoroughly documented in the literature.

All statistical analyses were corrected for ApoE4 status, age, and gender, and results were corrected for multiple comparisons using family-wise error correction (p < 0.05) and threshold-free cluster enhancement.”

  1. It is advised to include ‘‘*’’ when significant results are obtained in the table. The reviewer was not able to find in the manuscript the degrees of freedom as well as the confidence intervals.

 We have updated Table 2 to reflect the differences between groups as requested. We added the individual comparisons p-values.

TBI-/CDR≥0.5

TBI-/CDR=0

TBI+/CDR≥0.5

TBI+/CDR=0

TBI-/CDR≥0.5 vs. TBI-/CDR=0

TBI-/CDR≥0.5 vs. TBI+/CDR≥0.5

TBI-/CDR=0 vs. TBI+/CDR=0

TBI+/CDR≥0.5 vs. TBI+/CDR=0

# of participants

N=50

N=50

N=10

N=10

Gender:

1

0.01

1

0.065

 Female

24 (48.0%)

25 (50.0%)

0 (0.00%)

5 (50.0%)

 Male

26 (52.0%)

25 (50.0%)

10 (100%)

5 (50.0%)

APOE4 Status:

1

1

1

1

 Negative

30 (60.0%)

32 (64.0%)

5 (50.0%)

7 (70.0%)

 Positive

20 (40.0%)

18 (36.0%)

5 (50.0%)

3 (30.0%)

Age

75.1 (7.82)

76.8 (6.30)

76.6 (7.63)

75.4 (7.31)

0.61

0.93

0.94

0.982

TBI Type:

.

.

.

0.65

 Concussion

7 (70.0%)

5 (50.0%)

Head Injury

3 (30.0%)

5 (50.0%)

TBI Source:

.

.

.

1

Accident

1 (20.0%)

0 (0.00%)

Fall

2 (40.0%)

2 (100%)

Football

1 (20.0%)

0 (0.00%)

MVA

1 (20.0%)

0 (0.00%)

ADAS Total

10.2 (3.96)

8.74 (3.68)

13.5 (4.01)

8.07 (2.01)

0.217

0.05*

0.95

0.007**

ADAS-Cog

16.0 (6.48)

12.7 (5.89)

20.1 (6.34)

11.9 (3.32)

0.037

0.2

0.98

0.01**

MOCA

23.9 (3.49)

25.8 (2.92)

21.5 (4.20)

26.7 (1.58)

0.022

0.15

0.87

0.004**

MMSE

28.2 (2.06)

28.9 (1.47)

26.4 (3.03)

28.6 (0.97)

0.182

0.03*

0.96

0.047*

GD Total

1.84 (2.16)

0.84 (0.96)

2.50 (2.12)

0.78 (0.83)

0.017

0.66

1

0.12

CDR Memory

0.53 (0.12)

0.00 (0.00)

0.65 (0.58)

0.00 (0.00)

0

0.25

0.84

<0.001***

CDR GLOBAL

0.50 (0.00)

0.00 (0.00)

0.50 (0.33)

0.00 (0.00)

0

1

1

<0.001***

Every Day Cognitive Assessment

Memory

2.35 (0.72)

1.62 (0.50)

2.20 (0.72)

1.71 (0.57)

<0.001

0.89

0.98

0.33

Language

2.02 (0.70)

1.45 (0.41)

1.91 (0.71)

1.32 (0.31)

<0.001

0.95

0.92

0.12

 Visual spatial

1.49 (0.59)

1.11 (0.20)

1.50 (0.56)

1.06 (0.12)

<0.001

1

0.99

0.13

 Planning

1.57 (0.63)

1.12 (0.26)

1.54 (0.57)

1.11 (0.23)

<0.001

0.99

1

0.21

 Organization

1.75 (0.79)

1.15 (0.24)

1.58 (0.50)

1.24 (0.52)

<0.001

0.84

0.97

0.56

 Divided Attention

2.19 (0.86)

1.45 (0.56)

1.82 (0.75)

1.25 (0.25)

<0.001

0.44

0.87

0.29

 Total

1.90 (0.58)

1.32 (0.26)

1.76 (0.49)

1.28 (0.28)

<0.001

0.81

0.99

0.09

FAQ Total

2.52 (3.80)

0.14 (0.50)

5.90 (5.99)

0.25 (0.46)

0.001

0.009**

1

0.001***

Clock drawing tests

4.52 (1.01)

4.78 (0.51)

4.30 (1.06)

4.60 (0.70)

0.394

0.86

0.92

0.84

Clock copy test

4.86 (0.50)

4.82 (0.48)

4.70 (0.67)

4.50 (1.58)

0.992

0.90

0.50

0.91

Logic memory - Story

11.3 (4.74)

15.2 (3.81)

10.1 (4.04)

15.3 (2.54)

<0.001

0.83

1

0.03

Logic memory- Delayed Recall

9.40 (4.88)

14.0 (3.37)

6.80 (3.94)

15.1 (3.51)

<0.001

0.27

0.88

<0.001***

Category Fluency Test

18.4 (5.19)

21.7 (4.56)

15.6 (5.08)

20.2 (4.37)

0.005

0.36

0.80

0.16

Trail Making Test

 Part A - Time to Complete (sec)

39.1 (25.3)

31.0 (8.16)

40.1 (12.1)

31.1 (9.48)

0.113

1

1

0.69

 Part A - Errors of Commission

0.12 (0.39)

0.12 (0.39)

0.11 (0.33)

0.00 (0.00)

1

1

0.77

0.91

 Part A - Errors of Omission

0.26 (1.84)

0.00 (0.00)

0.00 (0.00)

0.00 (0.00)

0.707

0.93

1

1

 Part B - Time to Complete (sec)

103 (66.7)

68.6 (27.8)

155 (90.0)

74.4 (27.4)

0.011

0.04*

0.99

0.008**

 Part B - Errors of Commission

0.69 (1.06)

0.37 (0.64)

1.89 (1.76)

0.20 (0.42)

0.32

0.004**

0.96

0.001**

 Part B - Errors of Omission

0.35 (1.55)

0.02 (0.14)

1.40 (2.32)

0.00 (0.00)

0.536

0.06

1

0.05*

Trail Making Test

 Forgetting

3.47 (5.75)

3.88 (2.85)

5.33 (2.50)

4.29 (3.45)

0.975

0.76

1

0.97

 Immediate

36.6 (11.0)

44.6 (10.5)

28.2 (6.59)

48.6 (7.52)

0.005

0.26

0.79

0.004***

 Learning

5.18 (3.41)

5.81 (2.26)

2.83 (1.94)

6.29 (1.80)

0.737

0.22

0.97

0.11

 Percent Forgetting

41.6 (80.8)

36.5 (33.4)

79.0 (23.6)

36.6 (31.1)

0.978

0.45

1

0.54

 Recognition Score

11.2 (2.61)

12.6 (3.12)

9.40 (4.35)

13.7 (0.87)

0.072

0.29

0.77

0.01**

  1. A table with only the results of logistic regression should be provided.

I didn’t understand what logistic regression the reviewer means. We did not run a logistic regression analysis in this work.

  1. What were the exclusion criteria? Were the individuals in the use of any medication?

Unfortunately, we don’t have information about the participants’  medication status, but are bound by the constraints of the ADNI database. We add the following observation to the limitations section: “Furthermore, we note that this study relies upon retrospective self-reporting of TBI, without objective clinical details; this limitation of the database necessarily imparts some uncertainty about the nature and severity of the reported injury [15]. The lack of detailed clinical information on self-reported TBI, unmedically-defined severity level of the TBI, and history of medications use are also key limitations of the study, which might have contributed to the molecular profiles of these participants.”

References:

  1. Mohamed, A.Z.; Nestor, P.J.; Cumming, P.; Nasrallah, F.A. Traumatic Brain Injury Fast-Forwards Alzheimer’s Pathology: Evidence from Amyloid Positron Emission Tomorgraphy Imaging. J. Neurol. 2022, 269, 873–884, doi:10.1007/s00415-021-10669-5.
  2. Morris, J.C. The Clinical Dementia Rating (CDR): Current Version and Scoring Rules. Neurology 1993, 43, 2412–2414, doi:10.1212/wnl.43.11.2412-a.
  3. Folstein, M.F.; Folstein, S.E.; McHugh, P.R. “Mini-Mental State”. A Practical Method for Grading the Cognitive State of Patients for the Clinician. J. Psychiatr. Res. 1975, 12, 189–198, doi:10.1016/0022-3956(75)90026-6.
  4. Nasreddine, Z.S.; Phillips, N.A.; Bédirian, V.; Charbonneau, S.; Whitehead, V.; Collin, I.; Cummings, J.L.; Chertkow, H. The Montreal Cognitive Assessment, MoCA: A Brief Screening Tool for Mild Cognitive Impairment. J. Am. Geriatr. Soc. 2005, 53, 695–699, doi:10.1111/j.1532-5415.2005.53221.x.
  5. Skinner, J.; Carvalho, J.O.; Potter, G.G.; Thames, A.; Zelinski, E.; Crane, P.K.; Gibbons, L.E. The Alzheimer’s Disease Assessment Scale-Cognitive-Plus (ADAS-Cog-Plus): An Expansion of the ADAS-Cog to Improve Responsiveness in MCI. Brain Imaging Behav. 2012, 6, 489–501, doi:10.1007/s11682-012-9166-3.
  6. Farias, S.T.; Mungas, D.; Reed, B.R.; Cahn-Weiner, D.; Jagust, W.; Baynes, K.; DeCarli, C. The Measurement of Everyday Cognition (ECog): Scale Development and Psychometric Properties. Neuropsychology 2008, 22, 531–544, doi:10.1037/0894-4105.22.4.531.
  7. Sheikh, J.I.; Yesavage, J.A. 9/Geriatric Depression Scale (Gds) Recent Evidence and Development of a Shorter Version. Clin. Gerontol. 1986, 5, 165–173, doi:10.1300/J018v05n01_09.
  8. Ito, K.; Hutmacher, M.M.; Corrigan, B.W. Modeling of Functional Assessment Questionnaire (FAQ) as Continuous Bounded Data from the ADNI Database. J. Pharmacokinet. Pharmacodyn. 2012, 39, 601–618, doi:10.1007/s10928-012-9271-3.
  9. Shulman, K.I. Clock-Drawing: Is It the Ideal Cognitive Screening Test? Int. J. Geriatr. Psychiatry 2000, 15, 548–561, doi:10.1002/1099-1166(200006)15:6<548::AID-GPS242>3.0.CO;2-U.
  10. Vakil, E.; Blachstein, H. Rey Auditory‐verbal Learning Test: Structure Analysis. J. Clin. Psychol. 1993, 49, 883–890, doi:10.1002/1097-4679(199311)49:6<883::AID-JCLP2270490616>3.0.CO;2-6.
  11. Barr, A.; Brandt, J. Word-List Generation Deficits in Dementia. J. Clin. Exp. Neuropsychol. 1996, 18, 810–822, doi:10.1080/01688639608408304.
  12. Arnett, J.A.; Labovitz, S.S. Effect of Physical Layout in Performance of the Trail Making Test. Psychol. Assess. 1995, 7, 220–221, doi:10.1037/1040-3590.7.2.220.
  13. Williams, B.W.; Mack, W.; Henderson, V.W. Boston Naming Test in Alzheimer’s Disease. Neuropsychologia 1989, 27, 1073–1079, doi:10.1016/0028-3932(89)90186-3.
  14. McGurn, B.; Starr, J.M.; Topfer, J.A.; Pattie, A.; Whiteman, M.C.; Lemmon, H.A.; Whalley, L.J.; Deary, I.J. Pronunciation of Irregular Words Is Preserved in Dementia, Validating Premorbid IQ Estimation. Neurology 2004, 62, 1184–1186, doi:10.1212/01.WNL.0000103169.80910.8B.
  15. Corrigan, J.D.; Bogner, J. Initial Reliability and Validity of the Ohio State University TBI Identification Method. J. Head Trauma Rehabil. 2007, 22, 318–329, doi:10.1097/01.HTR.0000300227.67748.77.

Round 2

Reviewer 1 Report

Retrospective studies, while extremely useful, come with extreme limitations. It is unfortunate that the authors have such limited clinical detail when attempting to make such assertions. Nevertheless, I believe the authors have adequately relayed to the reader that their findings require the (missing) context of a full clinical picture and TBI Hx. Accept in present form. 

Reviewer 2 Report

Satisfactory.

The reviewer would like to provide some considerations.

1. IRB number for the majority of the IRB and journals should be specific. A clear statement in the study project should be done. Publishing more than one paper with the same IRB number is characterized as ''salami science''.

Hoit JD. Salami science. Am J Speech Lang Pathol. 2007 May;16(2):94. doi: 10.1044/1058-0360(2007/013). PMID: 17456887.

2. A specific copyright permission for each test is mandatory.

https://blogs.ucl.ac.uk/copyright/2017/11/17/psychometric-scales-copyright-protection-and-translation/#:~:text=Q.,required%20to%20reuse%20it%20lawfully.

The authors should hold these two requirements.